# Interannual wave-driven shoreline change on the California coast

William C. O'Reilly [1] ✉, Mark A. Merrifield[1] ✉, Laura Cagigal[2], Dayeon Yoon[1], Holden Leslie-Bole[1], Susheel Adusumilli[3], Adam P. Young[1], K. Vos[4] & R. T. Guza[1]

The important role of wave climate variability in driving shoreline evolution has been demonstrated recently with improved satellite-derived shoreline detection algorithms, wave buoy records, and wave reanalysis and hindcast models. While severe beach erosion with extreme El Niño waves is well documented on Pacific coastlines, less clear is the broader link between interannual wave energy and shoreline response. Here, we show half of California's interannual Landsat shoreline change is a coherent response to wave power anomalies originating from a specific central North Pacific swell generation region, which in turn is only weakly correlated with the Niño3.4 index. Positive wave power anomalies (beach narrowing) are strongly associated with El Niños, but the negative anomalies (beach widening) are not similarly tied to La Niñas. The North Pacific wave climate modulation of beach width narrowing and widening over interannual to multi-decadal time scales has implications for long-term coastal resilience planning.

The empirical relationship between beach shoreline changes and incident wave energy or energy flux has been studied across many time and space scales. In California (CA), erosion occurs in response to energetic winter waves during strong El Niños[1–6]. Comprehensive analyses[7,8] of teleconnections between the El Niño Southern Oscillation (ENSO), wave power anomalies, and shoreline change have further linked the wave climate to observed interannual cycles of both beach narrowing and widening throughout the Pacific Basin, using Landsat satellite-derived shorelines[9] (CoastSat) and the fifth generation European Center for Medium-Range Weather Forecasts (ECMWF) wave reanalysis (ERA5)[10]. CoastSat has been used to examine long-term CA shoreline evolution[11], which is characterized by alternating longshore reaches of shoreline narrowing and widening, or beach rotations, the result of decadal variations in wave-driven longshore transport[12] restricted by geologic (headlands) and engineered features (harbors, large jetties).

Climate variations in North Pacific wave energy are closely linked to the intensity and position of the Aleutian Low[13]. On interannual time scales, El Niño events associate with an eastward-shifted Aleutian Low, increasing the frequency of extreme wave events in the central North Pacific, which in turn influences shoreline change trends along the California coast[14–16]. A similar pattern also occurs on decadal time scales

with the positive (warm) phase of the Pacific Decadal Oscillation (PDO)[17]. Here, we build upon this previous work by focusing on correlated interannual shoreline and wave power changes on the CA coastline.

Underlying CA's geographically complex seasonal (<1 yr) shoreline variations[18,19], and superimposed on decadal (»1 yr) longshore rotation trends[11] (red and blue lines, Fig. 1d), is strikingly coherent statewide interannual cross-shore beach widening and narrowing (black lines, Fig. 1c, d).

In this work Landsat shoreline positions (Fig. 2) and ERA5 reanalysis wave power (Fig. 3) are reduced to annual means (October through September) (Fig. 4, Supplementary Fig. S5) to locate the source of the statewide interannual shoreline coverability. Coherent time-longshore shoreline behavior is isolated using empirical orthogonal function analysis (EOF, Fig. 5, Supplementary Fig. S2) and linked to the North Pacific wave climate with ESTELA[20] (Evaluation of Source and Travel-time of wave Energy reaching a Local Area, Fig. 6).

## Results

### ‹CAcoastSat› and offshore wave power validation
Tide corrected Landsat shoreline positions spanning 37 years[9] (1985–2021, CoastSat v1.2) are reduced to seasonally weighted annual

[1]Scripps Institution of Oceanography, UC San Diego, La Jolla, CA, USA. [2]Universidad de Cantabria, Santander, Spain. [3]University of Oregon, Eugene, OR, USA. [4]OHB Digital Services, Bremen, Germany. ✉e-mail: woreilly@ucsd.edu; mamerrifield@ucsd.edu

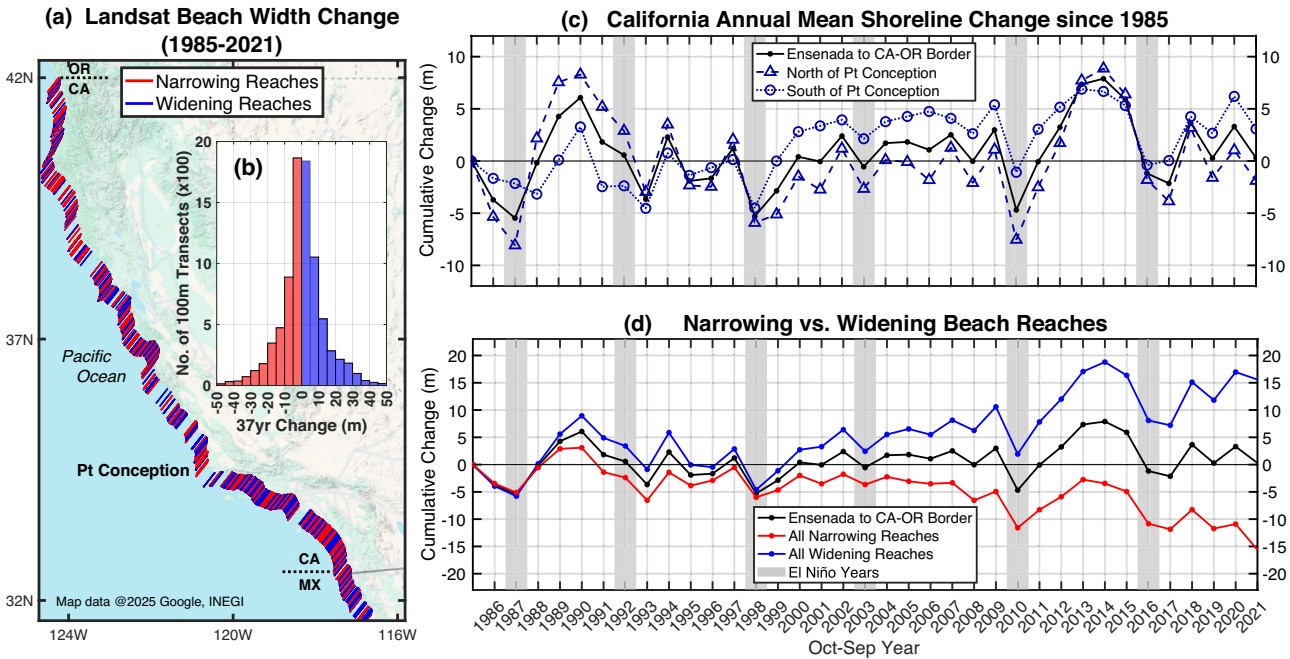

**Fig. 1 | Summary of 1985–2021 interannual shoreline evolution from Ensenada, MX to the California-Oregon border (850 km), based on annual (Oct–Sep) means at 8468, 100m-spaced Landsat transects[9]. a** Longshore reaches of net beach narrowing (red) and widening (blue). **b** Markedly symmetric distribution of transects based on their net narrowing/widening magnitude. **c** Regionally averaged interannual shoreline evolution versus time. **d** Average evolution of narrowing/widening transects (relative to statewide mean) versus time.

means (October through September), and averaged in the longshore over contiguous beach reaches, to reduce errors and isolate interannual change (Fig. 2a, hereafter <CAcoastSat>). Solid curves are seasonally-weighted, longshore-averaged, annual mean shoreline positions relative to their respective long-term means (see Methods). This approach minimizes artifacts from image digitization, wave runup, and irregular sampling. <CAcoastSat> agrees with similarly averaged annual mean, mean high water (MHW) shoreline elevation location anomalies (<Survey MHW>) obtained from in situ surveys at five beaches in San Diego County[21–23] (RMSE = 2.5 m – 4.5 m, $r^2$ = 0.70 – 0.93, Fig. 2b) with varying lengths (span of longshore-averaged transects) of 1–3 km. Annual mean shoreline changes from Landsat and surveys have similar El Niño year erosion (2010, 2016), interannual variability, and localized widening from sand nourishments at Imperial Beach, Cardiff, and Solana Beach starting in 2012-13. Coherent multi-year shoreline beach change patterns at many CA beaches are apparent using annual and beach-scale longshore averaging <CAcoastSat > (Fig. 2c).

Buoy observations show the CA offshore annual wave power climate increases 25% from south to north in the mean (Fig. 3b), but the interannual variation of the power anomaly, and therefore annual wave power change, is spatially homogeneous ($r^2$ = 0.86, Fig. 3c). Offshore wave power from the ECMWF ERA5 wave reanalysis (i.e. ERA5 vector wave energy flux, units m³/s) offshore of south-central CA is validated against 2001–2021 Datawell Directional Waverider deep water buoy observations[24] ($r^2$ = 0.88, see Methods). North Pacific winter swell (peak direction 300° and peak frequency 0.07 Hz) dominate the offshore annual wave power anomaly variability (Fig. 3d) compared to Southern Ocean summer swell power, which is an order of magnitude weaker (peak direction 185° and peak frequency 0.065 Hz, Fig. 3e, note reduced wave power scales).

### Correlated interannual shoreline change and offshore wave power

A time-longshore diagram of <CAcoastSat> shoreline change, the difference between sequential annual means, illustrates coherent

statewide beach narrowing during El Niño years with large positive power changes (e.g. 1998, 2010, 2016, red years, Fig. 4a), and widening over negative power change periods between El Niños (e.g. 2011–2014, blue years). Shoreline change averaged statewide, and for northern CA (San Luis Obispo to Del Norte Counties, north of Pt. Conception) and southern CA (including a section of northern Baja California) emphasize the strikingly similar statewide response (Fig. 4b). Statewide patterns of <CAcoastSat> shoreline change are inversely correlated with wave power change ($r^2$ = 0.72, Fig. 4b). This aligns with broader findings that ENSO-driven wave climate anomalies significantly alter wave energy flux and drive shoreline erosion[4,25]. North-south differences are most evident prior to the 1997-98 El Niño.

To highlight variability owing to beach rotations, the 37-yr <CAcoastSat> dataset is divided into transects with cumulative widening or narrowing since 1985 (Figs. 1b and 4c). The cumulative mean change associated with the near-equal number of narrowing (red line, 51%) and widening (blue line, 49%) transects is substantial (+/−15 m, Fig. 4c) and consistent with littoral cell-scale changes[11]. When viewed as statewide means, the present era of beach rotations within the littoral cells appears to have started in earnest after the 1997-98 El Niño, or at the onset of the current PDO cold phase, specified using climatic North Pacific wave power[15] (black arrow, Fig. 4c).

The symmetric distributions of transects that have cumulatively eroded or accreted (Fig. 1b) essentially cancel each other out in the statewide mean (black line, Fig. 4c). We anticipated a negative 1985–2021 trend owing to the reduction of sand supply from rivers and increased armoring of coastal cliffs[26–29]. Instead, the Landsat dataset indicates that the CA-wide annual mean shoreline position has varied roughly +/−7m since 1985, with a trend that is not significantly different than zero (dotted black line, Fig. 4c). We explore this result, along with the unexpectedly coherent component of statewide shoreline recovery, with an EOF analysis of the <CAcoastSat> shoreline change data (Fig. 4a).

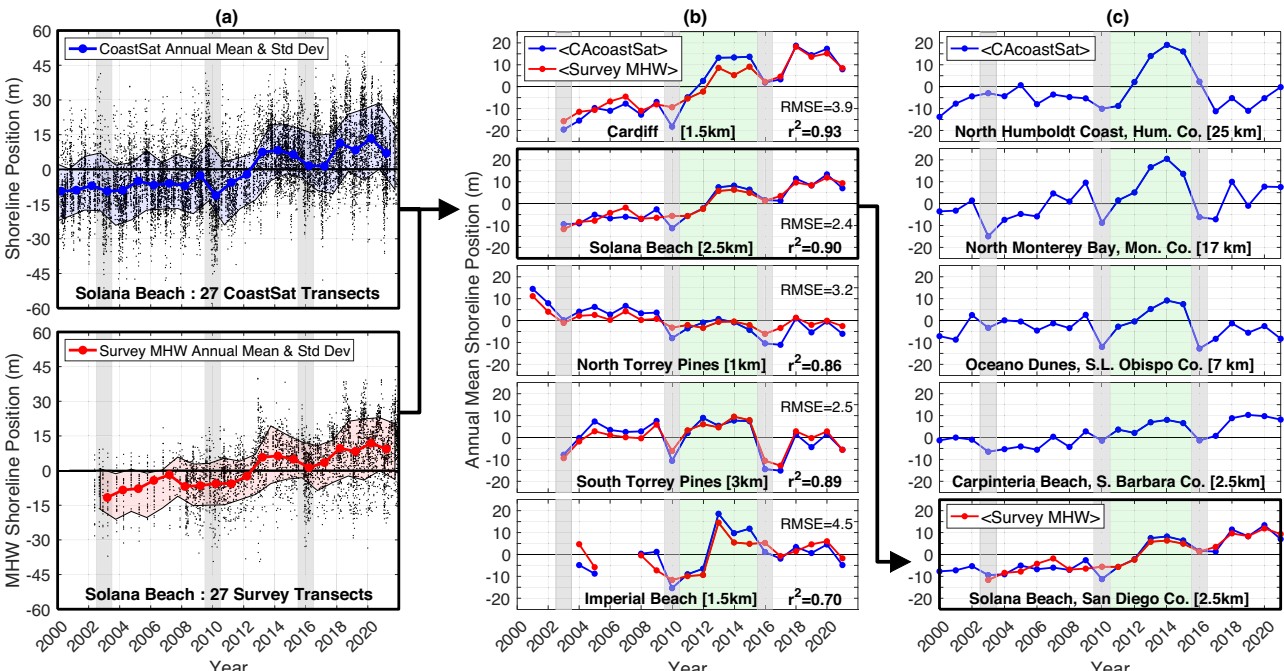

**Fig. 2 | Illustration of <CAcoastSat> derivation, validation, and application to CA beaches. a** Shoreline position at Solana Beach (dots) versus time from CoastSat (upper panel) and survey transects (lower panel). **b** <CAcoastSat> and <Survey MHW> time series comparisons at five San Diego County beaches (beach lengths in brackets). **c** <CAcoastSat> time series at five beaches spanning the State highlight the strong CA-wide beach recovery period (green shading) between the 2010 and 2016 El Niños (gray shaded years). Sites are north to south, top to bottom (see Counties in Fig. 3a).

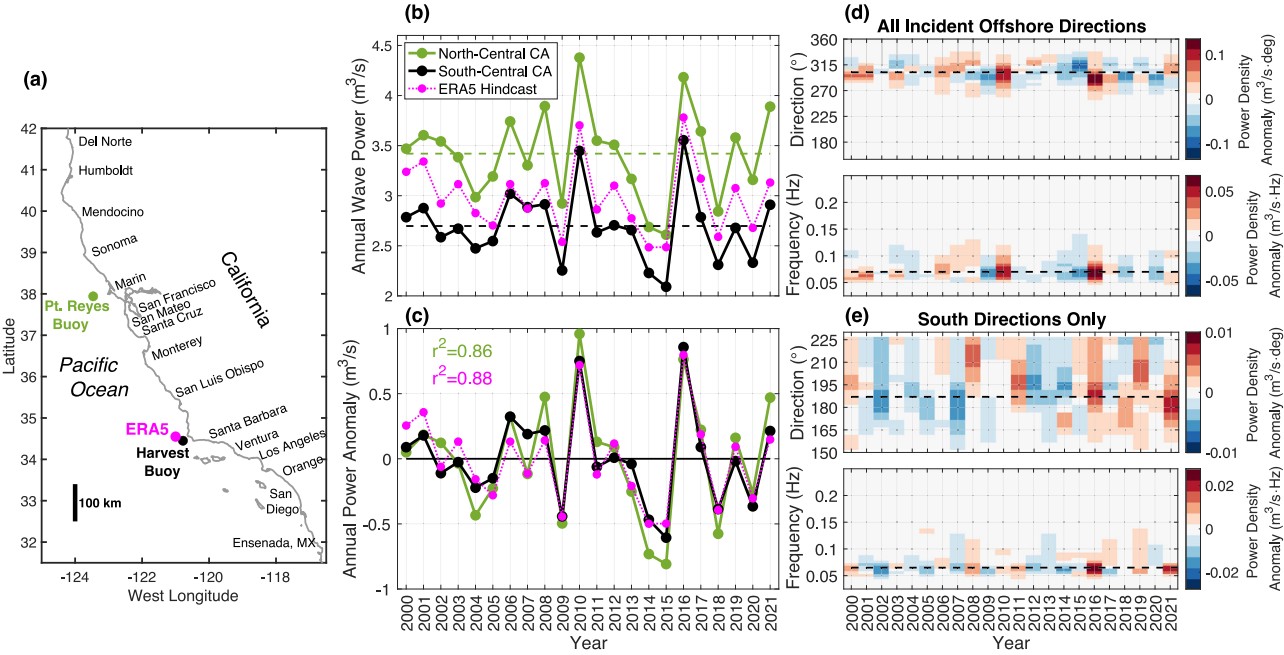

**Fig. 3 | Offshore ERA5 reanalysis wave power validation. a** Locations of observed north-central (Pt. Reyes Buoy), observed south-central (Harvest Buoy), and modeled south-central CA (ERA5) wave power. **b** Offshore annual mean wave power (i.e., energy flux, units m³/s) versus time; south-central CA reanalysis (ERA5, magenta) and observed with buoys in south-central (black) and north-central (green) CA. **c** Annual mean anomalies (relative to means) of north- and south-central CA. **d** Buoy Maximum Entropy Method (MEM) south-central CA offshore annual wave power density anomalies (color bar) versus (upper) wave direction and (lower) frequency. **e** Observed power anomalies with southern MEM directions. Dashed lines are 22-yr mean peak directions and frequencies.

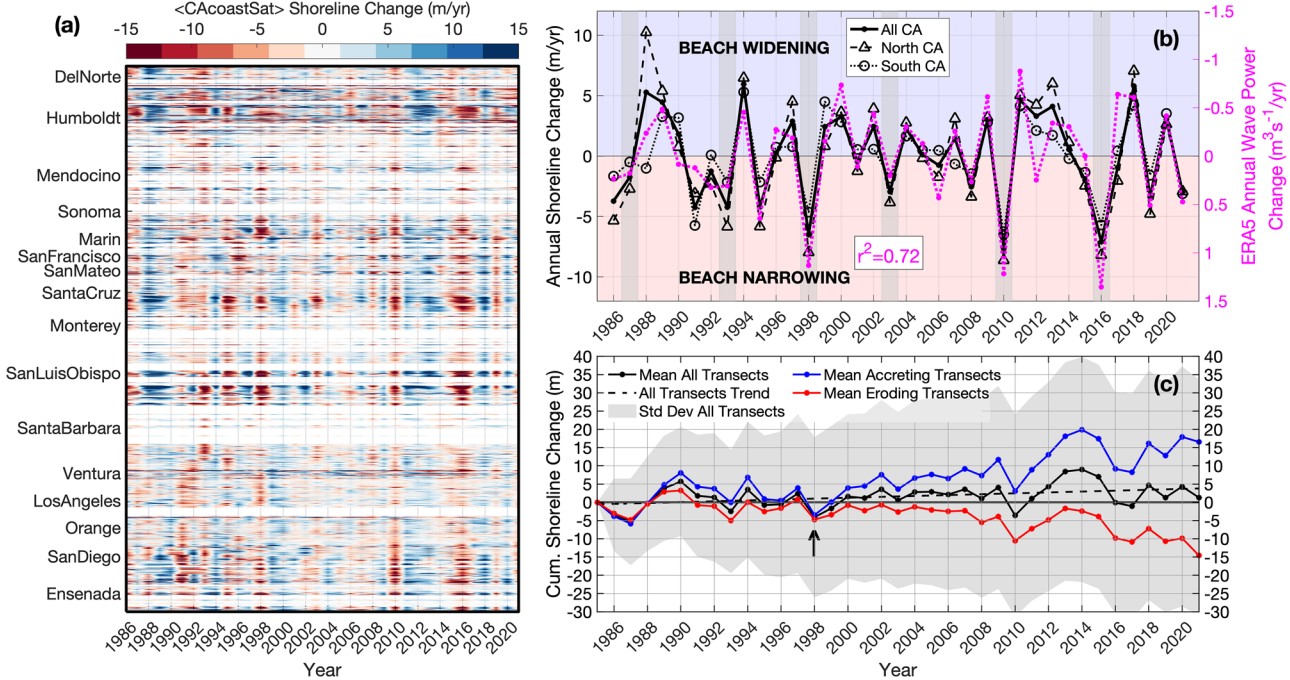

**Fig. 4 | CA shoreline and offshore wave power change. a** Annual shoreline change (color bar, difference of successive annual mean <CAcoastSat> shorelines) versus longshore location and time. Locations lacking sand beaches denoted by white horizontal stripes. **b** Annual change averaged over all CA, North CA (San Luis Obispo to Del Norte County) and South CA (Ensenada, Mexico to Santa Barbara County). El Niño years are shaded in gray. CA-wide shoreline change inversely correlates with south-central CA ERA5 annual offshore wave power change (purple, inverted right y-axis). **c** Cumulative change in shoreline position (since 1985) versus time for widening (blue, 4318 transects), narrowing (red, 4150 transects), and all (black, 8468 transects). Evolution of beaches into widening and narrowing transects (beach rotation) accelerated after the 1997-98 El Niño (black arrow). The statewide average 37-yr trend is small (black dashed line).

## Dominant modes of CA shoreline variability

Dominant statewide patterns of annually averaged CA shoreline change are identified with EOF analysis of time series with the time mean removed at each transect and a 5-km running mean (Fig. 5, Supplementary Fig. S2). The first two EOF modes account for 50% and 7% of the total shoreline change variance. The mode 1 (Fig. 5a, Supplementary Fig. S2a) spatial pattern is mostly positive with considerable longshore variability, indicating in-phase covariations of varying amplitude. Mode 1 is notably weak in parts of Southern California (Fig. 5a), presumably due to wave shadowing caused by the shoreline orientation change south of Point Conception and sheltering by offshore islands[30,31]. Mode 1 is well-correlated with ERA5 ($r^2 = 0.76$), demonstrating that interannual shoreline variations are strongly correlated with wave power anomalies[32].

Mode 2 (Fig. 5b, Supplementary Fig. S2b) represents regional-scale deviations from the dominant statewide shoreline response. This mode is most energetic in the early Landsat record (1988–1992), when significant spatial differences in shoreline change occur between northern and southern California (Fig. 5e). The greater prominence of mode 2 in earlier years coincides with both strong ENSO variability and potential limitations in Landsat-derived shoreline resolution. Various measures of wave power variations across the state using the buoy data and ERA5 reanalysis were unable to account for mode 2 change. The mode 1 + 2 EOF residual (43% of variance, Fig. 5c, Supplementary Fig. S2c) contains changes on smaller littoral cell-scales including beach nourishments and longshore transport beach rotations (ref. 11 and Fig. 1a). When reduced to annual statewide means, only mode 1 has a significant interannual signal.

## Mode 1 shoreline regulated by central North Pacific swell generation

CA wave reanalysis backtracking with ESTELA[12,20] identifies the primary north and south Pacific source regions of below and above average wave power for different <CAcoastSat> mode 1 and ENSO phases (Fig. 6, see methods). We use the hourly wind sea wave partition from the Centre for Australian Weather and Climate Research (CAWCR) wave hindcast[33,34] to analyze relationships between the wave generation areas and shoreline change. The composite generation averages (Fig. 6) are based on the more significant positive and negative years for mode 1 (index over or below the 75th and 25th percentile, Fig. 6a, b), and the Niño3.4 index (Oct-Sep Niño3.4 index with > 5 months above/below 1/−1, Fig. 6c, d).

The correspondence of erosion with El Niño events is well known; however, we find only a weak overall correlation between shoreline change and the ENSO index ($r^2 = 0.21$), because non-El Niño years produce wave power anomalies with significant variability depending on ENSO-related SST anomaly distribution and intensity[35] (Fig. 6e). Higher correlation is found using just the January Niño3.4 index value each year ($r^2 = 0.32$) where individual El Niño years correlate well, but there is still poor correlation in the years between El Niños (2010–2016 is an exception, green shading, Fig. 6e). Below-normal northern hemisphere energy years (Fig. 6b), associated with mode 1 Landsat shoreline recovery, often occur during ENSO neutral years.

## Discussion

Analogous to seasonal beach change in CA[18,19], mode 1 represents the cross-shore transport and storage of sand on interannual time scales. During energetic wave years sand moves offshore and the shoreline narrows until subsequent wave conditions favor shoreward transport. The onshore-offshore exchange (mode 1) is coherent statewide and tied to the offshore wave climate, despite significant nearshore wave climate mean energy variation owing to island sheltering and coastal orientations, suggesting that offshore power anomalies lead to nearshore power anomalies of the same sign but different magnitudes. The larger/smaller self-selected grain sizes associated with the higher/lower mean energy beaches[36] would further contribute to more similar

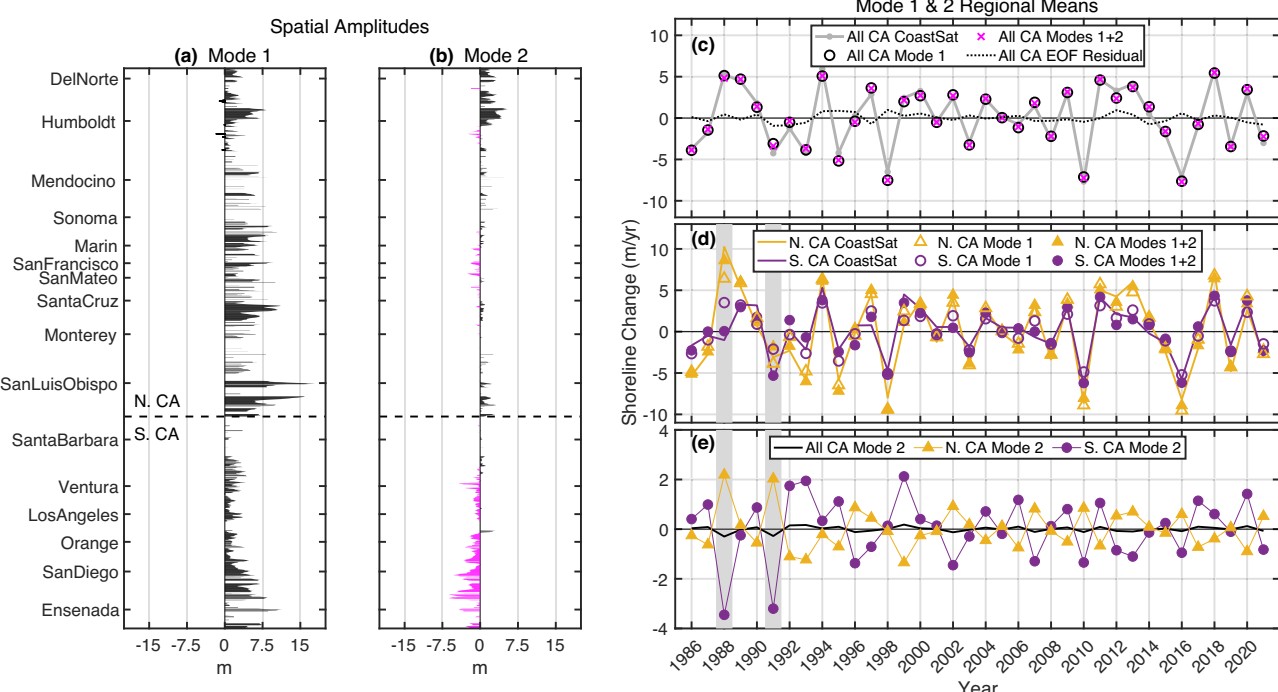

**Fig. 5 | EOF spatial amplitudes and regional means. a, b** Mode 1, 2 spatial amplitudes at 8460 longshore locations (5-km running mean) multiplied by the standard deviation of the modal temporal expansion. **c, d** Mode 1 captures coherent statewide mean <CAcoastSat> shoreline change. **e** Mode 2 represents change difference between N. and S. CA. Largest mode 2N/S differences occur early in the record (1988, 1991 gray shaded years). Jan 17-18, 1988 was the largest wave event in the ERA5 1940-2021 record and primarily impacted S. CA[52,53]. Mode 2 (panel **e**, black line) and EOF residual (panel **c**, black dotted line) statewide mean changes are small. Vertical scales differ in (**c**–**e**). Time-longshore plots of the two EOF modes and the residual are in Supplementary Fig. S2.

cross-shore transport rates and more coherent interannual beach narrowing and recovery. Additional analysis of <CAcoastSat> errors at the five Sand Diego beaches (Supplementary Fig. S5) suggests that the interannual <CAcoastSat> change signal is likely amplified by correlated annual mean wave runup anomalies in the (only tide corrected) shoreline position estimates, but have little impact on longer trends in the annual mean time series.

Changes at a specific beach or within a littoral cell often differ significantly from the statewide average. For example, shoreline width trends in San Diego County vary from beach to beach with some mirroring the statewide average more closely than others (Fig. 2b). The large residual changes at smaller littoral cell spatial scales (Supplementary Fig. S2c) reflect local longshore transport[11], sand supply[37] (including nourishments), and beach orientation[19] effects. Nevertheless, it is important that underlying any year-to-year shoreline change at a specific beach is a foundational statewide beach response to annual mean offshore wave power anomalies (Fig. 5a, Supplementary Fig. S2a) with a common source region in the central North Pacific (Fig. 6a, b).

These findings have direct implications for coastal management and long-term hazard mitigation along the CA coast. A shoreline change climatic index[12] based on the sign and strength of the central North Pacific winter west wind anomaly could provide annual guidance for CA shoreline managers in conjunction with seasonal to interannual statistical and machine learning models[38]. Anticipated beach recovery phases during neutral ENSO conditions could guide the timing of shoreline protection and restoration projects. Future work could further investigate the variability of recovery dynamics, given that swell energy propagating from the Southern Ocean is strongly positively correlated with the Southern Annular Mode, particularly during Northern Hemisphere summer months[39].

While our results indicate no significant net statewide shoreline narrowing over the past 37 years, the long-term wave-driven narrowing trend underscores the vulnerability of California's beaches to changing climatic conditions. The Landsat shoreline estimates begin in 1985, 8 years after the 1976-77 Pacific Ocean climate shift[40] and does not include strong El Niños in 1977-78 and 1982-83. A more speculative ERA5 83-yr shoreline change hindcast (Supplementary Fig. S1) estimates a statewide wave-driven narrowing of ~4 m since 1941 with possible Pacific Decadal Oscillation (PDO) connections. As sea level rise accelerates beach erosion through the end of the century[41], integrating interannual wave-driven shoreline behavior into resilience planning will be critical for protecting coastal infrastructure, ecosystems, and recreational spaces.

## Methods

### CoastSat shoreline location time series

The CoastSat[9] multi-decadal time series of shoreline positions greatly expands the historical database. Online CoastSat 1985–2021 shoreline position time series are determined by algorithmically detecting the instantaneous water-sand and/or predominant wet-dry sand interface in individual satellite images for coastal reaches with sand or gravel beaches. CoastSat Version v1.2 shoreline positions (e.g. upper panel, Fig. 2a) are tide-corrected relative to MSL (but not wave runup corrected) using a single, long-term mean beach slope estimated from the uncorrected position time series and estimated tide levels, and spatially averaged onto 100m-spaced cross-shore transects. CoastSat data are routinely updated with both new observations and improved analyses of historical images (https://zenodo.org/records/15614554). Version v1.2 was used to be consistent with previous publications relating CoastSat v1.2 to Pacific climate patterns. The most recent version of CoastSat (v1.6) yielded very similar results (see Supplementary Figs. S3, 4).

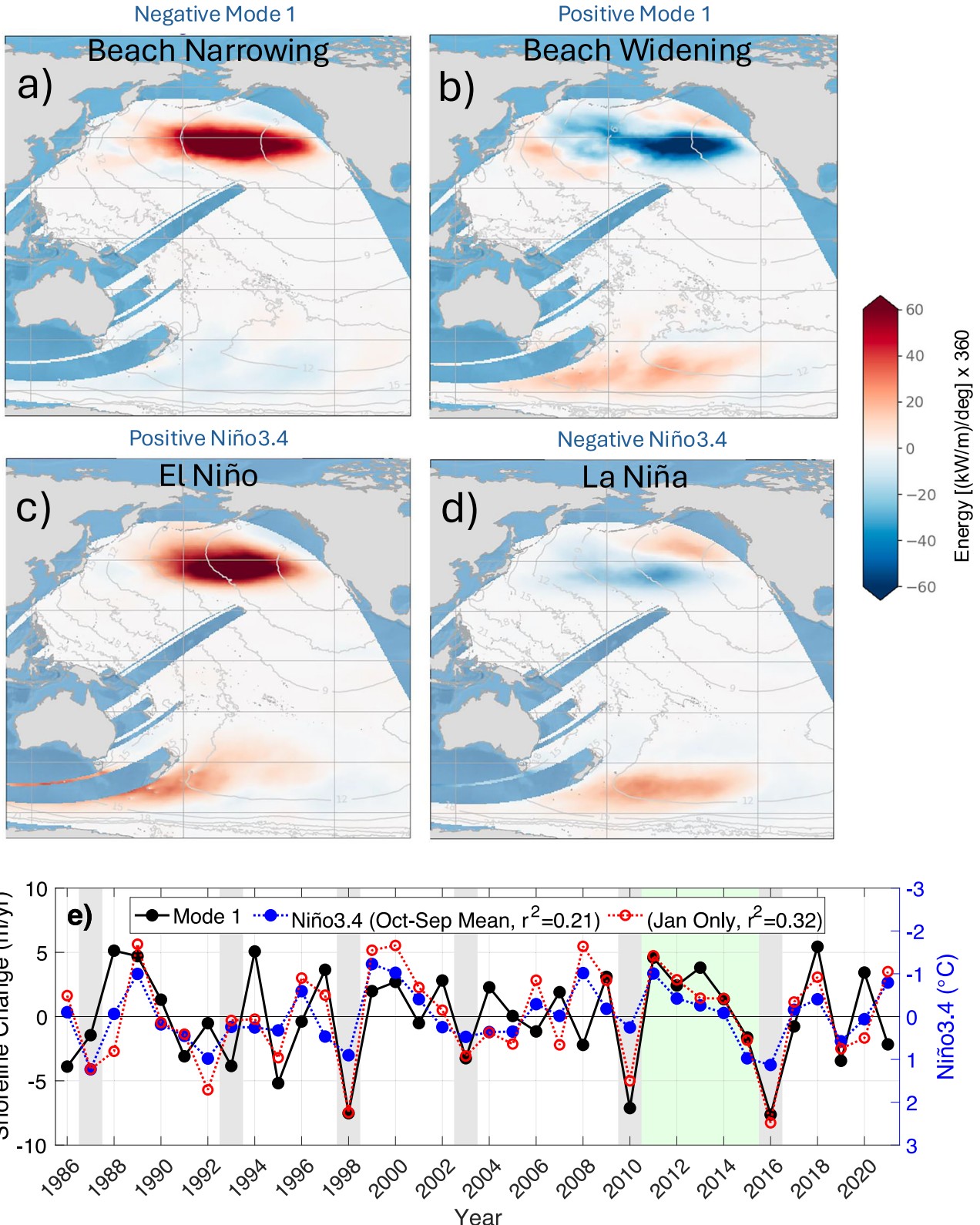

**Fig. 6 | ESTELA analysis of Pacific wave energy sources associated with EOF mode 1 shoreline change and the Niño3.4 index.** El Niño years are shaded gray. **a**, **b** Wave generation anomaly regions associated with the shoreline EOF mode 1 and **c**, **d** ENSO Niño3.4 index. **e** Correlations between mode 1 shoreline change and annual mean (blue, $r^2 = 0.21$) and mid-winter (red, $r^2 = 0.32$) Niño3.4 index are relatively low. In contrast, green shading highlights 2010−2016 with higher correlation during robust statewide beach recovery (Fig. 2c).

### Surveyed mean high water contour location time series

From 2002 to 2010, the San Diego County coastline was surveyed in spring and fall using airborne topographic LiDAR. From 2001 to 2021, various San Diego beaches were surveyed with ATV, truck-mounted Lidar and push dolly-mounted mobile GPS. The mobile GPS surveys overlap in time and space with both the airborne and mobile LiDAR data sets and have been used to ensure survey accuracy and consistency between data collection methods[22,42–46]. From 2017 to 2021, truck and ATV-mounted mobile LiDAR surveys were monthly or quarterly at many San Diego beaches.

The geographic location of CoastSat v1.2 MSL tide-corrected shorelines track approximately with the surveyed mean high water (MHW) elevation contour, possibly owing to wave runup[47]. Time series of the MHW contour location on 100m-spaced cross-shore transects[48] (MOP lines) were derived from the survey data (e.g. lower panel, Fig. 2a). The longshore locations of the MOP and CoastSat 100m-spaced transects are interlaced, but their orientations typically differ by less than a few degrees. The five San Diego County beaches with the most comprehensive coverage were used for CoastSat validation.

### Annual mean shoreline change estimation

To minimize biases owing to temporally irregular sampling and very local longshore variations, both survey MHW and CoastSat transect time series are averaged in time and longshore space (<CAcoastSat>, <Survey MHW>). First, the time series are reduced to annual means (October-September) by progressively time averaging to monthly, quarterly, semi-annual, and finally annual values. Second, the time series are further reduced to annual mean anomalies, by removing their respective long-term means, to minimize any spatial offsets between the CoastSat and survey (GPS) geographic reference frames (Fig. 2b). Finally, for the survey-CoastSat comparisons at each beach, a single space-time averaged annual shoreline anomaly time series is derived by spatially averaging over 10–30 transects depending on beach length.

For the statewide EOF analysis, a fixed 5 km (50 transect) longshore running mean is used. Based on the <CAcoastSat> and <Survey MHW> comparisons in San Diego County (Fig. 2, Supplementary Fig. S5), tide correction errors owing to beach slope variations, and any seasonal biases associated with the unresolved wave runup signal, are minimized in the annual averages. However, <CAcoastSat> annual errors are also correlated with wave power, suggesting the annual mean wave runup anomalies may be amplifying the <CAcoastSat> annual change signal by a few m, but appear to have little impact on cumulative change and trends (Supplementary Fig. S5).

### Empirical orthogonal function analysis of shoreline change data

At each grid point, a singular value decomposition (using svd.m in MATLAB) is applied to the Landsat annual shoreline change data, i.e., annual shoreline position first differenced in time. A temporal mean is removed from the first-differenced time series, equivalent to removing a linear trend from the shoreline position data. Each location is smoothed with a 5 km longshore running mean. The modal spatial patterns (Fig. 5a, b) are proportional to the temporal standard deviation of each mode.

### Offshore annual mean wave power anomaly estimates

Wave forcing (periods 2–40 s) is characterized as the offshore wave energy flux, or power (units m³/s), using Coastal Data Information Program (CDIP) directional wave buoy observations from 2000 to 2022[24]. Temporal fluctuations in the observed annual wave power (energy flux) anomalies covary along the CA offshore boundary. Mean wave power is 25% higher in north- than south-central CA (Fig. 3b) but annual mean anomalies (relative to means), and therefore annual wave power change, have similar magnitude and are well correlated (Fig. 3c).

Smaller positive wave power anomalies are associated with (summer) Southern Ocean swell (Fig. 3e).

Historical 2001–2021 annual mean offshore wave power anomalies are derived from a single continuous wave power record of blended deep-water buoy observations. CDIP buoys are used preferentially, followed by nearby NOAA buoys, with any remaining time gaps filled by long-term monthly means. For central and southern CA, 96% of the wave power time series is from the CDIP Harvest buoy station west of Pt Conception, with the limited data gaps filled by data from the nearby CDIP Harvest Southeast buoy, the San Nicolas Island buoy or nearby NOAA buoys. Remaining data gaps (-1%) were filled with the time series mean for the same time of year (30 day running average). For north-central CA, waves are mostly from the CDIP Pt Reyes buoy station (93%), with data gaps filled by CDIP Monterey Bay West buoy (4%), and with the CDIP Pt Sur, NOAA 46042 and 46013 buoys.

### The ECMWF ERA5 wave reanalysis annual mean wave power estimates

The wave ERA5 reanalysis[10] from the European Centre for Medium-Range Weather Forecast (ECMWF) provides hourly, global waves with 0.5° spatial resolution from 1940 to present. ERA5 significant wave height, $H_s$, and peak period, $T_p$, parameters are used to estimate the hourly offshore power (wave energy flux, $H_s g T_p / 64\pi$ m³/s) and annual (Oct-Sep mean) wave power anomalies. Harvest Buoy and ERA5 annual power anomalies, and therefore offshore wave power change, are significantly correlated ($r^2 = 0.88$, Fig. 3c).

### ESTELA analysis

The ESTELA (Evaluation of Source and Travel-time of wave Energy reaching a Local Area) method[20] estimates the source location and propagation time of waves approaching a specific location. Unlike previous studies that used both the sea and swell wave partitions[20,49,50], we use the hourly wind sea wave partition from the Centre for Australian Weather and Climate Research (CAWCR) wave hindcast[33,34] to better isolate wave generation regions.

To analyze the relationship between the wave generation areas and the shoreline response, composite averages are computed using the mode 1 index (over or below the 75th and 25th percentile). The anomaly of the energy generation with respect to the mean (Fig. 6a, b) displays a clear positive/negative anomaly in the northern hemisphere during the negative/positive phase of the index, i.e., more energetic waves in the generation region correspond to narrower CA shorelines and vice versa. A composite using Oct-Sep years when the Niño3.4 index was above/below 1/−1 °C for more the 5 months shows a similar pattern as mode 1 during El Niño years (Fig. 6c), but well below normal wave conditions do not generally occur during La Niña Years (Fig. 6d) leading to a poor overall correlation between mode 1 shoreline change and the Niño3.4 index.

## Data availability

The satellite position and wave data used in this study are available in the Code Ocean repository https://doi.org/10.24433/CO.3917025.v1.

## Code availability

The MATLAB code used to analyze the data and produce the figures in this study is available in the Code Ocean repository[51] https://doi.org/10.24433/CO.3917025.v1.

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

## Acknowledgements

This study was funded by the U.S. Army Corps of Engineers W912HZ1920020 (W.C.O., M.A.M., S.A., A.D.Y., R.T.G.), the California Department of Parks and Recreation, Natural Resources Division Oceanography Program C19E0026 (W.C.O., M.A.M., S.A., A.D.Y., R.T.G.), and the Office of Naval Research N00014-23-1-2170 (H.L.-B.). The comments of three referees helped improve the manuscript significantly.

## Author contributions

W.C.O., S.A. and H.L.-B. worked collaboratively on the extended CoastSat analysis and survey validation. M.A.M. and D.Y. worked collaboratively on the EOF analysis. L.C. provided the ESTELA analysis. M.A.M. developed the shoreline hindcast model. A.P.Y. and R.T.G. provided the survey data. K.V. created the baseline CoastSat data used in this study. W.C.O., M.A.M., L.C., D. Y., H.L.-B., S. A., A P.Y., K.V., and R. T. G. W.C.O., M.A.M., L.C., D. Y., H.L.-B., S.A., A.P.Y., K.V., and R.T.G. provided background information, intellectual contributions, writing, and/or editing of the manuscript.

## Competing interests

The authors declare no competing interests.
