## [Transparent Peer Review file · Nature Communications]

Interannual Wave-Driven Shoreline Change on the California Coast

Corresponding Author: Dr William O'Reilly

Version 0:

Reviewer comments:

Reviewer #1

(Remarks to the Author)

The manuscript explores the links between shoreline changes and wave climatology for the coastline of California. The topic has already been explored in other manuscripts focusing on climatic patterns effects on shoreline around the Pacific Ocean, although the analysis presented in this manuscript is much more detailed and insightful. The topic is complicated since the data is noisy and the processes driving shoreline change are poorly understood. The results presented show that beach change at the interannual scale is the result of a delicate balance modulated by the North Pacific wave climate. This type of study is relevant globally as it could potentially be similarly applied to other coastlines around the world and discover the possible wave climate drivers of beach change.

The analysis is properly described, and my only comment is related to the use of shoreline data extracted from satellite. Some of the shoreline changes are small and I wonder if a comment should be made about the resolution of the data and if it might affect the interpretation of the results.

I believe the authors should be a bit clearer when explaining the role of nourishments on the long-term shoreline trend. At lines 258-259 it is indicated that the estimate does not consider sand supply changes like nourishments. Does it mean that the very limited shoreline retreat would be much larger without nourishments? Notice that the abstract does not mention nourishments, so the reader has the false impression that California beaches are stable (and lines 258-259 are not particularly clear).

The authors do not touch on the implications of this study for long-term predictability of beach change, and I think they should. My reading of this paper is that if I wanted to predict the future of California beaches, I would need accurate predictions of the drivers of the wave climate (i.e., the Niño3.4 index), which at present is beyond our capability. This makes for a major source of uncertainty on future predictions.

I am certainly not an expert, but I believe the January Niño 3.4 Index is used as an indicator of El Niño strength and storminess. I was hoping the authors would discuss in more detail what is special about the January Niño 3.4 Index.

The analysis is sound, the results are of interest to the broad nearshore community, the manuscript is well written, and the figures are of high quality. I am happy to suggest publication of the manuscript pending the minor revisions above.

(Remarks on code availability)

Reviewer #2

(Remarks to the Author)

“Interannual Wave-Driven Shoreline Change on the California Coast”
By O'Reilly, Merrifield, et al.

This paper presents an analysis of satellite-derived shoreline data on the California Coast, focusing on relationships between the interannual variability of shoreline erosion and wave energy. The authors find clear links between the annual shoreline anomaly and the wave-energy anomaly across the state. Erosion/accretion anomalies are also well linked with

climate variability (ENSO, PDO) as well as the regional wave patterns that are most closely linked to erosion/recovery anomalies. All of the wave-shoreline relationships found in the paper are abundantly clear from the presented analyses, which makes the paper a worthy contribution to the literature. Although, the connection between waves and shoreline anomalies has been quite well known (e.g., Barnard et al., 2015, Vos et al., 2023 ... and other papers by the Scripps authors on this paper), I think the clarity of the relationships presented as well as some of the nuances of wave-shoreline response relationship discussed in the current paper (e.g., the storm pattern analysis and the modal analysis) make this paper worth publishing.

I do, however, recommend some minor revisions, since I think the paper could lean into the nuances just a bit more to emphasize how this paper extends what we already know about the relationship between waves and shoreline response. For example, I'd personally appreciate a bit more presentation/investigation on the Mode 2 shoreline response (What is it? What does it look like it does? What wave patterns are associated with it?). To me, Mode 2 looks like a strong Southern California/Baja regional shoreline response that is muted when you go further north. Is this mode associated with a southern shift in the extratropical storms or a response to enhanced tropical storm activity? (since tropical storms are much stronger in the southern region compared to the north). The authors said they did not find a connection between this mode and the wave hindcasts, but I wonder how well those hindcasts resolve tropical storms (the buoys probably would detect the TC swells fairly well tho). It would be really cool to see a wave connection with the Mode 2 response if one exists.

I'm a bit worried that the authors find a strong Mode 2 response during the early Landsat years before Landsat 7 was available starting in ~1999. I would be interested to see if Mode 2 is still prominent in the analysis if you only include the post 1999 shoreline data. In working with satellite shoreline personally, I find post-Landsat 7 launch data set to be much more regular/reliable. If Mode 2 is found to be a bit of an artefact of the sparse early shoreline data then perhaps it would be appropriate to remove this Mode 2 in figure 5, and then the paper could focus purely on Mode 1. In short, I think Mode 2 is worth investigating a bit more in revisions.

Three more things:

1. You may consider inviting Kilian Vos, the creator of the shoreline data sets, to be a co-author on the paper. He put a lot of work into creating and curating updated versions of these data sets, which are central to the paper.

2. Kilian has released updated version of these data, thru 2025, and using an improved tidal model for tidal correction of the shoreline data. Those data are available online at: <https://zenodo.org/records/14889613> I know it would be a pain to use the updated data to extend your analysis from 2021 to the present, but you at least may consider using the Zenodo DOI for a better data reference in the data availability statement than just the website <http://coastsat.wrl.unsw.edu.au/>

3. You may consider keeping an eye out for a few in-press papers that are quite relevant to the work presented here (although this paper remains sufficiently different and valuable on its own, in my opinion). Marcan Graffin et al., "Waterline responses to climate forcing along the North American West Coast" in-press at Communications Earth & Environment. And "Are equilibrium shoreline models just convolutions?" Sean Vitousek (i.e., me) et al., in-press at JGR-Earth Surface. Overall, I liked reading the paper and found the results to be very clear, convincing and interesting.

Best of luck with revisions.

More specific comments:

Figure 2B: No green shading in Figure 2b?

Figure 3: Is there a reason that ERA5 (pink) seems larger than the Harvest buoy (black?) even though they seem almost co-located?

Figure 5: It would be really interesting to see what Mode 2 looks like similar to the plot of Mode 1 in a (I'm guessing large-scale cell rotations given the shape in panel c), perhaps this could be added to a supplemental?

You also may consider checking the units of the plots? Does one expect the units of panels b/c multiplied by the units of d/e/f to give the units of a?

Figure 6: It would be interesting to see the wave source regions for Mode 2, and if there's anything there related to the TC or bomb cyclone regions that affect SoCal more than NorCal, since Mode 2 seems to be a shape related to large changes/impacts in SoCal based on Figure 5c

Figure 6a,b,c,d: Units of "Energy" on colorbar are percentages here, right?

(Remarks on code availability)

Reviewer #3

(Remarks to the Author)

The manuscript "Interannual wave-driven shoreline change on the California coast" presents a regional-scale analysis of the main interannual drivers of shoreline change along this coastline combining satellite-derived shorelines and ERA5 waves. The work is very interesting, clearly written and generally rigorous. It builds on previous work looking at the drivers of shoreline change in California, with the main novelty in my opinion being the identification (through EOF and) of coherent shoreline patterns across the entire California coastline and linked to specific wave generating regions via ESTELA. I had some clarifications about the work however:

1) Curiously there is no mention of interannual fluctuations in sea-level anomalies driving shoreline change on this coastline. Several researchers (e.g. Barnard et al., 2015; Vos et al., 2023) suggested this could be a possible control of interannual shoreline change on the California coastline, with water levels also elevated during El Nino events. It would have been interesting to see how sea-level anomalies (e.g. using altimeter data) possibly correlate with Mode 1 and 2 Shoreline change as well - it might be that this also explains some of the variability?

2) Although interesting, I find the section linking long-term changes via ERA5 reanalysis, the PDO and shoreline a bit problematic. As found by Timmermanns et al., (2020, <https://doi.org/10.1029/2019GL086880>), reanalysis products like ERA5 should be treated with some scepticism in terms of long-term wave trends. Changes in data assimilation -particularly the transition between the pre-satellite era (1940-1979) to the satellite era (1979-) can create spurious trends in reanalysis data that might not be real (e.g. see Krueger et al 2013 for 20C reanalysis and Uppala et al., 2005 for ERA-40). This transition to the satellite era coincides with phase shifts in the PDO (Figure 7), so it is difficult to make any substantial claims here as to whether these patterns are meaningful. Likewise the subsequent modelling of shoreline change based on linear relationships is interesting but also very speculative – particularly when suggesting suggests "a statewide wave-driven retreat of ~ 4m loss since 1941". I would suggest that these results are downplayed a lot more.

3) I find the use of the term "beach loss" a bit problematic because it clearly shows the beach recovering and not lost. Perhaps "beach narrowing" would be better terminology throughout

4) How are individual beach reaches (e.g. each individual line in Figure 1a) defined (i.e. what determines them to be individual littoral cells)? Presumably this might influence the alongshore averaging and hence the EOF analysis?

Minor points:

Lines 52-55. This sentence is confusing as I thought it was referring to Figure 1 (I couldn't see any ERA5 or ESTELA analysis in the Figure) but I presume it is summarising the overall manuscript. This could be clarified better

Line 193. Figure 5 caption. I think you mean here "EOF mode 1 of Figure 4a" (not Figure 3a)

Acknowledgements section. Given the importance of the CoastSat shorelines to this work, an acknowledgement of Vos might be considered

(Remarks on code availability)

Code looks sufficient to replicate the analysis

Version 1:

Reviewer comments:

Reviewer #1

(Remarks to the Author)

The authors have addressed all comments appropriately and I appreciate the decision to include Vo as a co-author.

I see no reason to delay publication of this research.

(Remarks on code availability)

Reviewer #3

(Remarks to the Author)

I have read the revised manuscript "Interannual wave-driven shoreline change on the California coast", having reviewed the initial version. Overall I am satisfied with the amendments made and thank the authors for their careful consideration of the comments. I therefore think that the manuscript is suitable for publication in Nature Communications. I do have some very minor edits that I think can be addressed prior to publication:

Line 14: "increasing satellite imagery". It is not necessarily the increasing satellite imagery that has demonstrated this (as they have been collected steadily since the 1980s), but the algorithms. Suggest amend

Line 22 (and later on line 275): the word "retreat" is used here. Retreat is more a management response, so I suggest "narrowing" as more appropriate when referring to a reduction in beach width

Line 34 (and several other instances): "reanalysis wave hindcast". ERA5 is a wave reanalysis, not necessarily a wave hindcast (there are subtle differences). I suggest being consistent in referring it to the "ERA5 wave reanalysis"

Line 234: The use of the shortened "n34" for Nino3.4 is confusing - it took me a while to understand what this was and I suspect others will too. Some clarification of this might be warranted (as is the case for the title in Figure 6cd)

(Remarks on code availability)

REVIEWER COMMENTS

We appreciate the insightful comments of the 3 reviewers. Our response (red) follows each reviewer comment.

Reviewer #1 (Remarks to the Author):

The manuscript explores the links between shoreline changes and wave climatology for the coastline of California. The topic has already been explored in other manuscripts focusing on climatic patterns effects on shoreline around the Pacific Ocean, although the analysis presented in this manuscript is much more detailed and insightful. The topic is complicated since the data is noisy and the processes driving shoreline change are poorly understood. The results presented show that beach change at the interannual scale is the result of a delicate balance modulated by the North Pacific wave climate. This type of study is relevant globally as it could potentially be similarly applied to other coastlines around the world and discover the possible wave climate drivers of beach change.

The analysis is properly described, and my only comment is related to the use of shoreline data extracted from satellite. Some of the shoreline changes are small and I wonder if a comment should be made about the resolution of the data and if it might affect the interpretation of the results.

To address potential errors in annual mean shorelines from satellites more fully, we have added an additional supplemental section validation figure (Figure S5) and associated text describing how the errors may affect our interpretation of the shoreline response to changes in wave power (lines 333-336, and 696-701).

I believe the authors should be a bit clearer when explaining the role of nourishments on the long-term shoreline trend. At lines 258-259 it is indicated that the estimate does not consider sand supply changes like nourishments. Does it mean that the very limited shoreline retreat would be much larger without nourishments? Notice that the abstract does not mention nourishments, so the reader has the false impression that California beaches are stable (and lines 258-259 are not particularly clear).

Given the speculative nature of the 82-yr hindcast, the modeled long-term trend is now in the new supplemental section (lines 597-644 and Fig S1). To provide a better sense of how the nourishments would influence the EOF analysis, we have added an additional supplemental EOF figure (Figure S2) and some text in the main section of the paper (lines 181-183) regarding nourishments as being part of the EOF residual signal (Figure panel S2c), which represents 43% of the statewide variance. While nourishments are undeniably impactful on the shorelines that receive sand, and can be seen in the satellite shoreline data, they are lost in the large scale regional averaging used here to identify coherent changes (For CA's 800km of sand/gravel beaches, the cumulative length of unnourished beaches vastly exceeds the length of O(few km each) nourished ones from 1985-2021). Perhaps this would not be true for the 1940's-1960's time period when many S. CA harbors were constructed.

The authors do not touch on the implications of this study for long-term predictability of beach change, and I think they should. My reading of this paper is that if I wanted to predict the future of California beaches, I would need accurate predictions of the drivers of the wave climate (i.e., the Niño3.4 index), which at present is beyond our capability. This makes for a major source of uncertainty on future predictions. I am certainly not an expert, but I believe the January Niño 3.4 Index is used as an indicator of El Niño strength and storminess. I was hoping the authors would discuss in more detail what is special about the January Niño 3.4 Index.

We found that the Niño 3.4 index is not a good predictor of annual change in general, beyond strong El Niños leading to significant narrowing. Our results suggest that a new climate index based specifically on the strength of winds in the central north pacific would be needed to improve CA beach change predictability. Our findings indicate that long-term predictability is mostly dependent on any long-term trend in central north pacific wave power and sea level rise. We speak more directly to these points in a new dedicated discussion section at the end of the paper (lines 239-281).

The analysis is sound, the results are of interest to the broad nearshore community, the manuscript is well written, and the figures are of high quality. I am happy to suggest publication of the manuscript pending the minor revisions above.

Reviewer #2 (Remarks to the Author):

“Interannual Wave-Driven Shoreline Change on the California Coast”
By O’Reilly, Merrifield, et al.

This paper presents an analysis of satellite-derived shoreline data on the California Coast, focusing on relationships between the interannual variability of shoreline erosion and wave energy. The authors find clear links between the annual shoreline anomaly and the wave-energy anomaly across the state. Erosion/accretion anomalies are also well linked with climate variability (ENSO, PDO) as well as the regional wave patterns that are most closely linked to erosion/recovery anomalies. All of the wave-shoreline relationships found in the paper are abundantly clear from the presented analyses, which makes the paper a worthy contribution to the literature. Although, the connection between waves and shoreline anomalies has been quite well known (e.g., Barnard et al., 2015, Vos et al., 2023 ... and other papers by the Scripps authors on this paper), I think the clarity of the relationships presented as well as some of the nuances of wave-shoreline response relationship discussed in the current paper (e.g., the storm pattern analysis and the modal analysis) make this paper worth publishing.

I do, however, recommend some minor revisions, since I think the paper could lean into the nuances just a bit more to emphasize how this paper extends what we already know about the relationship between waves and shoreline response. For example, I’d personally appreciate a bit more presentation/investigation on the Mode 2 shoreline response (What is it? What does it look like it does? What wave patterns are associated with it?). To me, Mode 2 looks like a strong Southern California/Baja regional shoreline response that is muted when you go further north. Is this mode associated with a southern shift in the extratropical storms or a response to enhanced tropical storm activity? (since tropical storms are much stronger in the southern region compared to the north). The authors said they did not find a connection between this mode and the wave hindcasts, but I wonder how well those hindcasts resolve tropical storms (the buoys probably would detect the TC swells fairly well though). It would be really cool to see a wave connection with the Mode 2 response if one exists.

I’m a bit worried that the authors find a strong Mode 2 response during the early Landsat years before Landsat 7 was available starting in ~1999. I would be interested to see if Mode 2 is still prominent in the analysis if you only include the post 1999 shoreline data. In working with satellite shoreline personally, I find post-Landsat 7 launch data set to be much more regular/reliable. If Mode 2 is found to be a bit of an artefact of the sparse early shoreline data then perhaps it would be appropriate to remove this Mode 2 in figure 5, and then the paper could focus purely on Mode 1. In short, I think Mode 2 is worth investigating a bit more in revisions.

We have expanded Mode 2 and EOF residual details with a revised Figure 5, a new supplemental Figure S2 and an expanded explanation (lines 175-184). We agree our previous “partial inclusion of mode 2 and residual information” left the reader wanting. It seems more interesting now, if not completely satisfying, with Mode 2 and the residual fully included in the 2 figures and main narrative.

Three more things:

1. You may consider inviting Kilian Vos, the creator of the shoreline data sets, to be a co-author on the paper. He put a lot of work into creating and curating updated versions of these data sets, which are central to the paper.

Agree. Vos is now a co-author.

2. Kilian has released updated version of these data, thru 2025, and using an improved tidal model for tidal correction of the shoreline data. Those data are available online at: <https://zenodo.org/records/14889613> I know it would be a pain to use the updated data to extend your analysis from 2021 to the present, but you at least may consider using the Zenodo DOI for a better data reference in the data availability statement than just the website <http://coastsat.wrl.unsw.edu.au/>

Done. Latest CoastSat v1.6 results are now in supplementary section (lines 661-673, Figures S3,S4). Change to results are small.

3. You may consider keeping an eye out for a few in-press papers that are quite relevant to the work presented here (although this paper remains sufficiently different and valuable on its own, in my opinion). Marcan Graffin et al., “Waterline responses to climate forcing along the North American West Coast” in-press at Communications Earth & Environment. And “Are equilibrium shoreline models just convolutions?” Sean Vitousek (i.e., me) et al., in-press at JGR-Earth Surface.

Thanks. Graffin et al is indeed very related and has been added to introduction references.

Overall, I liked reading the paper and found the results to be very clear, convincing and interesting.

Best of luck with revisions.

More specific comments:

Figure 2B: No green shading in Figure 2b?

Added green shading to 2b.

Figure 3: Is there a reason that ERA5 (pink) seems larger than the Harvest buoy (black?) even though they seem almost co-located?

ERA5 total wave power is biased a little high at Pt Conception according to the Harvest Datawell Buoy. The buoy station is in fully exposed deep water, so likely not a sheltering issue. The ERA5 wave power is calculated from ERA5 wave parameter time series. The buoy wave power uses the full frequency spectra, so that may explain some of it. The power anomalies (i.e., bias removed) track very well (Fig 3c) which is really what we wanted, so we did not pursue the bias issue further.

Figure 5: It would be really interesting to see what Mode 2 looks like similar to the plot of Mode 1 in a (I'm guessing large-scale cell rotations given the shape in panel c), perhaps this could be added to a supplemental?

Mode 2 and Mode 1+2 residual Hovmollers are included as a new supplemental section Figure S2.

You also may consider checking the units of the plots? Does one expect the units of panels b/c multiplied by the units of d/e/f to give the units of a?

Figure 5a and Figure 5d/e/f (now Figure 5c/d/e) depict reconstructions of the CoastSat shoreline based on modes 1 and 2. These plots are in physical units. Figure 5b/c (now Figure 5a/b) show the spatial amplitude of modes 1 and 2 multiplied by the standard deviation of the mode temporal amplitude, which also are in physical units. We did not present the EOFs separated into temporal and spatial modes, in which case the expectation of the reviewer would have applied. We've tried to make this clearer in the caption of new Figures 5 and S2.

Figure 6: It would be interesting to see the wave source regions for Mode 2, and if there's anything there related to the TC or bomb cyclone regions that affect SoCal more than NorCal, since Mode 2 seems to be a shape related to large changes/impacts in SoCal based on Figure 5c

The ESTELA results for Mode 2 were inconclusive.

Figure 6a,b,c,d: Units of "Energy" on colorbar are percentages here, right?

ESTELA Units are $[(kW/m)/deg] \times 360$ and have been added to Figure 6.

Reviewer #3 (Remarks to the Author):

The manuscript “Interannual wave-driven shoreline change on the California coast” presents a regional-scale analysis of the main interannual drivers of shoreline change along this coastline combining satellite-derived shorelines and ERA5 waves. The work is very interesting, clearly written and generally rigorous. It builds on previous work looking at the drivers of shoreline change in California, with the main novelty in my opinion being the identification (through EOF and) of coherent shoreline patterns across the entire California coastline and linked to specific wave generating regions via ESTELA. I had some clarifications about the work however:

1) Curiously there is no mention of interannual fluctuations in sea-level anomalies driving shoreline change on this coastline. Several researchers (e.g. Barnard et al., 2015; Vos et al., 2023) suggested this could be a possible control of interannual shoreline change on the California coastline, with water levels also elevated during El Nino events. It would have been interesting to see how sea-level anomalies (e.g. using altimeter data) possibly correlate with Mode 1 and 2 Shoreline change as well - it might be that this also explains some of the variability?

We found low correlations between sea level anomalies and modes 1 and 2.

2) Although interesting, I find the section linking long-term changes via ERA5 reanalysis, the PDO and shoreline a bit problematic. As found by Timmermans et al., (2020, <https://doi.org/10.1029/2019GL086880>), reanalysis products like ERA5 should be treated with some skepticism in terms of long-term wave trends. Changes in data assimilation - particularly the transition between the pre-satellite era (1940-1979) to the satellite era (1979-) can create spurious trends in reanalysis data that might not be real (e.g. see Krueger et al 2013 for 20C reanalysis and Uppala et al., 2005 for ERA-40). This transition to the satellite era coincides with phase shifts in the PDO (Figure 7), so it is difficult to make any substantial claims here as to whether these patterns are meaningful. Likewise, the subsequent modelling of shoreline change based on linear relationships is interesting but also very speculative – particularly when suggesting suggests “a statewide wave-driven retreat of ~ 4m loss since 1941”. I would suggest that these results are downplayed a lot more.

Skepticism duly noted. The model has been moved to a new supplementary section. It is mentioned in the final discussion but described as “speculative” (lines 278-280).

3) I find the use of the term “beach loss” a bit problematic because it clearly shows the beach recovering and not lost. Perhaps “beach narrowing” would be better terminology throughout

Agree. Losses changed to narrowing throughout the paper.

4) How are individual beach reaches (e.g. each individual line in Figure 1a) defined (i.e. what determines them to be individual littoral cells)? Presumably this might influence the alongshore averaging and hence the EOF analysis?

The lines in Fig. 1a are generated directly from the annual mean CoastSat data from each 100m spaced transect based on whether their relative shoreline position was wider/narrower in 2021 than 1985. It is somewhat schematic, to get the reader to appreciate there are many rotation cells along the coast. It only shows rotations (widening/narrowing sign change) on alongshore length scales of 5km or longer (algorithm progresses up the coast looking for the next widening/narrowing sign change at least 5km away). This made the figure visually readable given the size of CA and consistent with the EOF analysis that applied a 5km moving mean to the CoastSat data matrix before calling the MATLAB singular value decomposition function.

Minor points:

Lines 52-55. This sentence is confusing as I thought it was referring to Figure 1 (I couldn't see any ERA5 or ESTELA analysis in the Figure) but I presume it is summarizing the overall manuscript. This could be clarified better

Made/expanded the sentence into its own paragraph summarizing the paper's topics (lines 60-65).

Line 193. Figure 5 caption. I think you mean here “EOF mode 1 of Figure 4a” (not Figure 3a)

Yes, Fig 5 caption has been revised and is now associated with a new supplemental Figure S2.

Acknowledgements section. Given the importance of the CoastSat shorelines to this work, an acknowledgement of Vos might be considered

Vos has been added as a co-author

Reviewer #3 (Remarks on code availability):

Code looks sufficient to replicate the analysis

REVIEWERS' COMMENTS

Reviewer #3 (Remarks to the Author):

I have read the revised manuscript "Interannual wave-driven shoreline change on the California coast", having reviewed the initial version. Overall I am satisfied with the amendments made and thank the authors for their careful consideration of the comments. I therefore think that the manuscript is suitable for publication in Nature Communications. I do have some very minor edits that I think can be addressed prior to publication:

Line 14: "increasing satellite imagery". It is not necessarily the increasing satellite imagery that has demonstrated this (as they have been collected steadily since the 1980s), but the algorithms. Suggest amend

Our admittedly ambiguous use of the word "increasing" was in reference to the notable increase in the number of successful shoreline algorithm detections from images each year beginning with Landsat 7 in 1999. We agree that even this advancement was secondary to the overall development of improved shoreline detection algorithms and have amended the sentence accordingly.

Line 22 (and later on line 275): the word "retreat" is used here. Retreat is more a management response, so I suggest "narrowing" as more appropriate when referring to a reduction in beach width

Agree. Replaced retreat with narrowing in all cases.

Line 34 (and several other instances): "reanalysis wave hindcast". ERA5 is a wave reanalysis, not necessarily a wave hindcast (there are subtle differences). I suggest being consistent in referring it to the "ERA5 wave reanalysis"

We removed the use of the word "hindcast" in all cases when referring to the ERA5 reanalysis.

Line 234: The use of the shortened "n34" for Nino3.4 is confusing - it took me a while to understand what this was and I suspect others will too. Some clarification of this might be warranted (as is the case for the title in Figure 6cd).

Yes, this was unnecessarily confusing on our part. n34 has been replaced with Nino3.4 in the text and Figure 6.